# Association between Circulating Levels of 25-Hydroxyvitamin D_3_ and Matrix Metalloproteinase-10 (MMP-10) in Patients with Type 2 Diabetes

**DOI:** 10.3390/nu14173484

**Published:** 2022-08-24

**Authors:** Daria Abasheva, Marta M. Dolcet-Negre, María A. Fernández-Seara, José María Mora-Gutiérrez, Josune Orbe, Francisco Javier Escalada, Nuria Garcia-Fernandez

**Affiliations:** 1School of Medicine, University of Navarra, 31009 Pamplona, Spain; 2Radiology Department, Clínica Universidad de Navarra, 31008 Pamplona, Spain; 3Instituto de Investigación Sanitaria de Navarra (IdiSNA), 31008 Pamplona, Spain; 4Nephrology Department, Clínica Universidad de Navarra, 31008 Pamplona, Spain; 5Laboratory of Atherothrombosis, Program of Cardiovascular Diseases, CIMA, University of Navarra, 31008 Pamplona, Spain; 6RICORS-Cerebrovascular Diseases, Instituto de Salud Carlos III, 28029 Madrid, Spain; 7Endocrinology and Nutrition Department, Clínica Universidad de Navarra, 31008 Pamplona, Spain; 8CIBER Fisiopatología de la Obesidad y Nutrición (CIBEROBN), Instituto de Salud Carlos III, 28029 Madrid, Spain; 9Red de Investigación Renal (REDINREN) and RICORS2040, Instituto de Salud Carlos III, 28029 Madrid, Spain

**Keywords:** 25-hydroxyvitamin D_3_, matrix metalloproteinase 10, Type 2 diabetes, diabetic kidney disease

## Abstract

Background: Matrix metalloproteinase-10 (MMP-10) levels increase progressively starting from early diabetic kidney disease (DKD) stages. Vitamin D_3_ (vitD_3_) deficit is associated with a higher risk of diabetic microangiopathy. Reduced MMP-10 expression has been observed after exposure to vitD_3_. Aim: to assess how vitD_3_ status is related to MMP-10 levels in patients with Type 2 diabetes (T2D). Methods: 256 patients with T2D were included in this cross-sectional study. Demographic, clinical and serum MMP-10 and 25-hydroxyvitamin D_3_ (25(OH)D_3_) levels were collected from each patient. The association between MMP-10 and (25(OH)D_3_) levels was assessed using a correlation analysis and fitting a multivariate linear regression model. Results: Serum MMP-10 levels were inversely correlated with circulating 25(OH)D_3_ (rho = −0.25; *p* < 0.001). In the subgroup analysis this correlation was significant in patients with DKD (rho = −0.28; *p* = 0.001) and in subjects with vitD_3_ deficit (rho = −0.24; *p* = 0.005). In the regression model adjusted for kidney function, body adiposity, smoking and vitD supplementation MMP-10 levels were 68.7 pg/mL lower in patients with 25(OH)D_3_ > 20 ng/mL, with respect to ≤20 ng/mL (*p* = 0.006). Conclusions: vitD_3_ repletion status is an independent predictor of MMP-10 levels in T2D patients. Perhaps, high 25(OH)D_3_ values should be targeted in these patients in order to prevent vascular complications.

## 1. Introduction

Altered expression of matrix metalloproteinases (MMPs) and tissue inhibitors of metalloproteinases (TIMPs) has been shown in diabetic kidney disease (DKD) [1,2]. These enzymes participate in the activation of growth factors and cytokines [3]. MMPs can be found in renal tissue [4]. Particularly, MMP-10 is expressed in the murine glomeruli and juxtaglomerular apparatus, and has been shown to degrade type II, IV and V collagens, gelatin and elastin [2]. Additionally, MMP-10 can activate other MMPs and inflammatory mediators, such as tumor necrosis factor alpha precursor (pro-TNF-α) [5]. In turn, MMP-10 expression is triggered by inflammatory stimuli present in chronic kidney disease (CKD), diabetes and atherosclerosis [6,7,8]. Renin–angiotensin–aldosterone system (RAAS) inhibitors prevented MMP-10 activation in a murine DKD model, suggesting that angiotensin II (Ang-II) could also activate a signaling pathway involved in MMP-10 expression regulation [8]. In patients with Type 2 diabetes (T2D) and CKD, elevated serum MMP-10 values were observed at early stages of nephropathy, followed by a progressive increase at more advanced CKD stages [8]. Furthermore, in patients with CKD serum, MMP-10 concentrations were associated with atherosclerosis severity [6].

The 25-hydroxyvitamin D_3_ (25(OH)D_3_) is the main circulating form of vitamin D_3_ (vitD_3_) and the most convenient biomarker of the vitD_3_ repletion status [9,10]. VitD_3_ is involved in multiple metabolic and immunological processes [11]. In patients with diabetes, a vitD_3_ deficit has been associated with a higher prevalence of microvascular complications [12]. Supplementation with vitD_3_ has been shown to improve systemic inflammation by reducing interleukin-6 and TNF-α levels [13]. Additionally, vitD_3_ has been shown to down-regulate the RAAS [14]. In CKD, the increase in Ang-II induces disintegrin and metalloproteinase-17 (ADAM17) expression, which activates the epidermal growth factor receptor/mitogen-activated protein kinase (EGFR-MAPK) pathway implicated in fibrosis, glomerulosclerosis, tubular dilation and atrophy [4]. VitD_3_ has been shown to inhibit ADAM17 expression and to block signal transmission in the EGFR–MAPK pathway [4]. Nevertheless, little is known about the relationship between vitD_3_ and other MMPs. We aimed to evaluate whether circulating 25(OH)D_3_ levels are associated with serum MMP-10 or TIMP-1 concentrations in patients with T2D.

## 2. Materials and Methods

### 2.1. Ethics

The study was approved by the University of Navarra Research Ethics Committee (2021.183TFG). Samples and patients’ data were provided by the University of Navarra Biobank and processed following standard operating procedures approved by the Research Ethics and Scientific Committees. All patients included in this study were properly informed and gave their written consent for sample collection, analysis and storage, as well as for use of their demographic and clinical data before inclusion.

### 2.2. Study Population

The original cohort included 271 patients with T2D recruited in 2009–2016 in the outpatient Endocrinology and Nephrology units of the Clinica Universidad de Navarra. The inclusion criteria were as follows: patients with T2D aged ≥18 years with or without CKD defined as an estimated glomerular filtration rate (eGFR) >60 mL/min/1.73 m^2^ and urine albumin–creatinine ratio (UACR) ≥30 mcg/mg or eGFR ≤60 mL/min/1.73 m^2^, independently of albuminuria. The exclusion criteria were: corticosteroid or immunosuppressive therapy, ongoing inflammatory, autoimmune or malignant process and/or data of probable non-diabetes-related nephropathy. For the final analysis, 256 patients were included, after eliminating participants with missing or extreme 25(OH)D_3_ (>100 ng/mL) values and patients with unknown T2D duration.

### 2.3. Data Acquisition and Measurement

All data was collected at the inclusion visit, comprising demographic (age and sex) and clinical (hypertension; cardiovascular diseases ((CVD), including ischemic heart disease, cerebrovascular ischemia and peripheral arterial disease; CKD; time since T2D diagnosis; weight and height), laboratory (fasting glycemia, serum cystatin C, creatinine, eGFR estimated by CKD-EPI 2021 formula [15], urea, urate, cholesterol (total and fractions), triglycerides, hemoglobin, calcium, phosphate, intact parathyroid hormone (PTH), C reactive protein (CRP), albumin, Hb1Ac and UACR) variables and information about treatments (loop diuretics, thiazides, RAAS inhibitors, paricalcitol and vitD supplements, including calcifediol and cholecalciferol). The MMP-10 and TIMP-1 serum levels were measured with the ELISA kit (DM1000 and DY970 respectively, R&D Systems, UK) according to the manufacturer’s instructions. The 25(OH)D_3_ levels were assessed using the immunoassay based on ElectroChemiLuminescence (ECL) technology. The month and the year of the sample acquisition were recorded to consider seasonal 25(OH)D_3_ variations. The body mass index (BMI) was calculated and the body fat percentage (BF%) was estimated using the Clinica Universidad de Navarra Body Adiposity Estimator (CUN-BAE), a previously validated formula based on age, sex and BMI [16,17]. Following the Endocrine Society Clinical Practice Guideline, the study population was divided into 3 groups according to the serum 25(OH)D_3_ level: normal vitD_3_ status (>30 ng/mL), insufficiency (20–30 ng/mL) and deficiency (<20 ng/mL) [9]. Alternative 25(OH)D_3_ cut-offs were also explored, based on national recommendations and considering the existing controversies regarding the optimal 25(OH)D_3_ level [18,19].

### 2.4. Statistical Analysis

The normality of distribution for continuous variables was checked using distribution graphs and the Shapiro–Wilk test. Qualitative data are presented as proportions; for quantitative data either a mean and standard deviation (SD) or a median and the interquartile range were calculated. The proportion of subjects with vitD_3_ deficit was compared in patients with and without CKD and with and without obesity using the Fisher’s exact test. To assess the correlations between MMP-10, TIMP-1 and 25(OH)D_3_ values, Spearman’s rank correlation coefficient was estimated, as these variables did not follow a normal distribution. The confidence intervals for the correlation coefficients were constructed using the bootstrapping procedure. The correlations between MMP-10, TIMP-1 and 25(OH)D_3_ values were evaluated in the entire sample, in different vitD_3_ status subgroups and in CKD/non-CKD patients. The mean values of logarithmically transformed MMP-10 were compared among the vitD_3_ categories using one-way analysis of variance (ANOVA), and post-hoc tests with the Bonferroni correction. For the comparison of median TIMP-1 and 25(OH)D_3_ values across vitD_3_ categories, the Kruskal–Wallis test was used. Then, a multivariate regression model was fitted using the bootstrapping procedure to address non-normal data distribution. 25(OH)D_3_ serum levels were first considered as a continuous and then as a categorical variable. The stepwise forward selection method was used to select covariates out of all the collected variables without missing values. The significance level was 0.05 for addition to the model and 0.2 for removal from the model. The final model was adjusted for the eGFR and BF% (variables identified by the stepwise analysis) and the smoking status and vitD supplementation (variables known to affect 25(OH)D_3_ values [10,20]). Two-tailed *p*-values < 0.05 were considered statistically significant. STATA 12.0 software was used for statistical analyses.

## 3. Results

Detailed characteristics of the patients stratified by the vitD_3_ status are given in Table 1. Overall, the majority of subjects in our study were male, except for the normal vitD_3_ status group, which had an equal proportion of men and women. The majority of participants had a relatively long-standing T2D, were non-smokers, overweight and suffered from hypertension. When compared with other groups, vitD_3_-deficient subjects showed a higher prevalence of hypertension and CKD and, consequently, tended to have higher serum creatinine and cystatin values and lower eGFR. In regard with medications, more vitD_3_-deficient subjects used RAAS inhibitors, compared to other groups. Finally, vitD_3_ supplementation was less prevalent and paricalcitol use was more prevalent among the vitD_3_-deficient subjects, although only a minority of patients were taking paricalcitol.

Overall, the prevalence of vitD_3_ deficiency (25(OH)D_3_ < 20 ng/mL) and vitD_3_ insufficiency (25(OH)D_3_ within 20–30 ng/mL) in our cohort was 50 and 25% respectively. The proportion of subjects with vitD_3_ deficit was significantly higher among patients with CKD, compared to the non-CKD group (61 vs. 40%, *p* = 0.001; Fisher exact test), and the median 25(OH)D_3_ was significantly higher in the non-CKD group (Figure 1). However, there was no difference between 25(OH)D_3_ levels in patients with different BMI or BF%. The proportion of subjects with vitD_3_ deficit was also similar among patients with and without obesity (*p* = 0.314 for BMI ≥ 30 kg/m^2^ vs. BMI < 30 kg/m^2^; *p* = 0.300 for BF% indicating obesity vs. lean/overweight subjects; Fisher’s exact test). There was no significant difference between median 25(OH)D_3_ values in samples obtained in winter (November to February) and those obtained in summer (*p* = 0.210; Wilcoxon rank-sum test).

Table 2 shows the median values and interquartile ranges of 25(OH)D_3_, MMP-10 and TIMP-1 in the whole sample and subgroups. MMP-10 levels were significantly different between subgroups, however, when the Bonferroni test for multiple comparison was conducted, only the vitD_3_-deficient subgroup showed a significant difference from both normal (*p* = 0.044) and insufficiency (*p* = 0.001) subgroups, while there was no difference in MMP-10 levels between normal and insufficiency subgroups (*p* = 0.808). TIMP-1 levels also significantly differed between subgroups, showing a progressive increase in subjects with an insufficiency or deficit of vitD_3_.

In the global correlation analysis, both MMP-10 and TIMP-1 showed a significant negative correlation with 25(OH)D_3_ levels (Figure 2 and Figure 3 respectively). When the correlation between MMP-10 and 25(OH)D_3_ was analyzed separately in patients with and without CKD, it remained significant only in the CKD subgroup. In the subgroup analysis by vitD_3_ categories, the correlation was significant only in the vitD_3_-deficiency subgroup (Table 3). However, when we explored higher 25(OH)D_3_ serum level cut-offs, corresponding to the 75th and 95th percentiles, we found a strong correlation between MMP-10 and 25(OH)D_3_ levels in patients with 25(OH)D_3_ levels >45 ng/mL and a moderate correlation in those with 25(OH)D_3_ levels within 30–45 ng/mL (Appendix A). TIMP-1 showed only a weak correlation with 25(OH)D_3_ globally, in patients with CKD and in those with avitD_3_ deficit, and remained similar applying different 25(OH)D_3_ cut-offs (Table 4). There was also a positive correlation between MMP-10 and TIMP-1 circulating levels (rho = 0.25, *p* < 0.001). The statistically significant results of the correlation analysis between other variables and 25(OH)D_3_ and MMP-10 levels are given in the Appendix A.

In the linear regression model, the 25(OH)D_3_ level used as a continuous variable was inversely associated with MMP-10 (beta-coefficient [95%CI] = −4.9 [−7.6; −2.2]); however, this association was no longer significant after adjusting for the eGFR (beta-coefficient [95%CI] = −2.2 [−4.4; 0.1]). Using the 20 ng/mL cut-off, we found a negative association between MMP-10 levels and vitD_3_ deficiency which remained significant after adjustment for the eGFR, body adiposity, smoking status and vitD supplementation. In the fully adjusted model, 25(OH)D_3_ level >20 ng/mL was associated with a 68.7 pg/mL reduction in the circulating MMP-10 concentration (Table 5). Using alternative cut-offs (30 ng/mL, 45 ng/mL), we did not observe any significant association between MMP-10 and 25(OH)D_3_ levels after adjusting for the eGFR.

## 4. Discussion

Our results suggest that circulating MMP-10 and TIMP-1 levels are inversely correlated with serum 25(OH)D_3_ levels in patients with T2D, particularly in those with a vitD_3_ deficiency and in those affected by CKD. Furthermore, 25(OH)D_3_ levels below 20 ng/mL are associated with higher circulating MMP-10 concentrations, independently of the eGFR.

The high prevalence of vitD_3_ deficiency and low median serum values of the 25(OH)D_3_ in our cohort are consistent with other studies of patients with diabetes [12,21]. We also found that 25(OH)D_3_ serum levels in our cohort were significantly lower in patients with CKD when compared to those without CKD. Although the 1,25(OH)_2_D deficiency in patients with kidney disease is biologically plausible, the deficit of 25(OH)D_3_ in this group of patients is less consistently reported. In the OSERCE study, which included CKD patients not undergoing dialysis, there was no correlation between 25(OH)D_3_ levels and kidney function, while a strong correlation was observed for calcitriol [22]. In contrast, another study reported that the progressive decline in 25(OH)D_3_ levels correlated with the eGFR decrease [23]. However, the former study excluded patients with nephrotic proteinuria, and included patients receiving vitD_3_ supplements, which could account for the lack of association between CKD stage and 25(OH)D_3_ concentration. The lower levels of the 25(OH)D_3_ can be explained by the loss of 25(OH)D_3_ bound to the vitamin D-binding protein in patients with nephrotic proteinuria, impaired ultraviolet B-dependent cutaneous synthesis of cholecalciferol and reduced hepatic 25(OH)D_3_ production due to uremia, dietary restrictions and less solar exposure in patients with CKD [24]. In our study, only 15% of patients had a UACR ≥ 300 and the median eGFR was relatively high. However, we did not collect information about solar exposure and diet in our patients.

This is the first study to show an association between the vitD_3_ status and MMP-10 circulating levels in humans. Elevated MMP-10 levels are found in various chronic inflammatory conditions and might play a particularly important role in diabetes as an early marker of developing microvascular complications (7,8). VitD_3_ is a known RAAS inhibitor [4] and in a mouse model of DKD; the pathological MMP-10 increase was blocked by an Ang-II receptor antagonist [8]. Furthermore, vitD_3_ has been shown to directly inhibit the cellular pathways involved in the activation of the MMPs transcription. *Bahar-Shany* et al. demonstrated that the inhibition of NF-kB cascade and Jun N-terminal kinase was responsible for the attenuation of the TNF-α-induced MMP-9 expression by calcitriol in a human keratinocyte culture [25]. Another study showed that pre-treatment of leukocytes infected with *M. tuberculosis* with calcitriol reduced both constitutive and induced MMP-10 expression, and reported a stimulation of prostaglandin E2 and interleukin-10 secretion by vitD_3_, both of which are known to suppress MMPs’ expression, elucidating yet another possible mechanism of vitD_3_-mediated MMPs’ inhibition [26]. In addition, calcitriol has been shown to inhibit the EGFR signaling pathway and reduce ADAM17 expression [4]. However, it has not been studied if the same mechanisms are involved in other MMPs’ regulation.

At the same time, the evidence regarding the association of vitD_3_ and MMPs in humans is scarce. A small study of African-American patients on hemodialysis found an inverse correlation between MMP-9 and 25(OH)D_3_ serum levels (r = −0.29), but not between MMP-2 and 25(OH)D_3_ levels. MMP-9 concentration was also significantly higher in patients with 25(OH)D_3_ < 15 ng/mL [27]. An interventional study of vitD_3_ supplementation in healthy adults reported a 68% reduction in MMP-9 and a 38% reduction in TIMP-1 levels following one year of vitD_3_ supplementation, however, only subjects with very low baseline 25(OH)D_3_ (<12 ng/mL) were included. Noteworthy, authors also studied the association between the vitamin D receptor (VDR) genotype and MMP-9 and TIMP-1 levels and showed that a specific allele of VDR gene was an independent predictor of TIMP-1 levels [28]. In contrast with these findings, a pilot study of the effect of 16-week vitD_3_ supplementation (50,000 UI per week) in subjects with metabolic syndrome showed no difference in serum MMP-2, MMP-9 and TIMP-1 concentrations between the intervention and placebo groups, although there was a significant reduction in the MMP-9 and TIMP-1 levels when compared to the baseline in the intervention group [29]. Although the aforementioned works point out that MMPs levels, and, therefore, vascular risk, increase particularly below the 12–15 ng/mL threshold in serum 25(OH)D_3_ concentration, we were able to observe this association at the 25(OH)D_3_ levels < 20 ng/mL. On the other hand, we showed that the strongest correlation between serum MMP-10 and 25(OH)D_3_ concentrations was observed at 25(OH)D_3_ levels > 45 ng/mL. Although this group of patients was rather small in our sample, it might indicate that higher 25(OH)D_3_ levels should be achieved in patients with diabetes to prevent vascular complications. In consistency with this assumption, a cross-sectional study of post-menopausal Spanish women showed no difference in glucose metabolism biomarkers between those with 25(OH)D_3_ > 30 ng/mL and those below this cut-off; however, in the group with 25(OH)D_3_ > 45 ng/mL there was a significant increase in insulin sensitivity and a decrease in basal insulin levels [30].

This study has several limitations. Firstly, due to the cross-sectional design, no causal inferences should be made. Secondly, although we restricted our study population to patients without any active inflammatory, autoimmune or malignant process, who were not taking immunosuppressants or steroids, and adjusted the regression model for some of the potential confounders, the residual confounding cannot be excluded. Thirdly, despite the relatively large sample size, our study might be underpowered to detect interactions between MMP-10 and 25(OH)D_3_ levels and CKD stage or eGFR, because the number of patients with moderate to severe CKD in our sample was low.

To conclude, we found that serum 25(OH)D_3_ concentration was significantly inversely associated with MMP-10 and TIMP-1 levels in patients with diabetes. VitD_3_ repletion status showed to be an independent predictor of serum MMP-10 values. These results suggest that high values of 25(OH)D_3_ should be targeted in patients with diabetes in order to decrease the risk of vascular complications. The efficacy of vitD_3_ supplementation in reducing circulating MMP-10 levels should be further examined in larger prospective studies.

## Figures and Tables

**Figure 1 nutrients-14-03484-f001:**
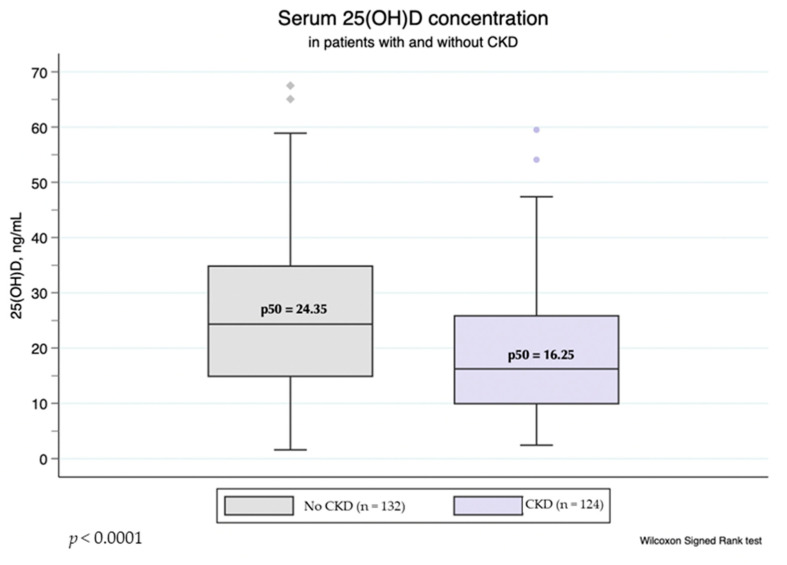
Serum 25(OH)D_3_ levels in patients with and without CKD.

**Figure 2 nutrients-14-03484-f002:**
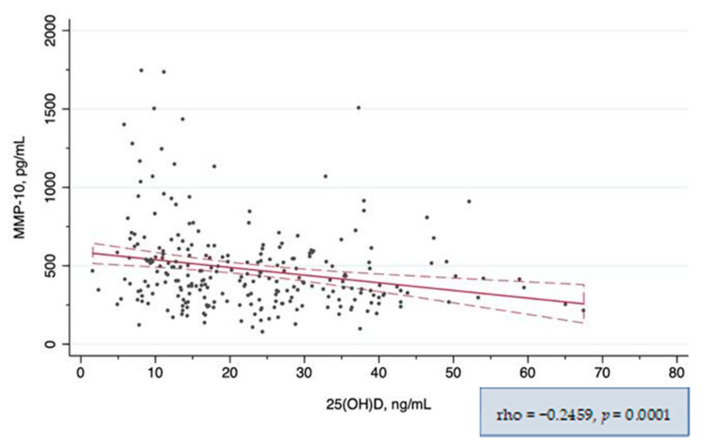
Correlation between serum MMP-10 and 25(OH)D_3_ levels.

**Figure 3 nutrients-14-03484-f003:**
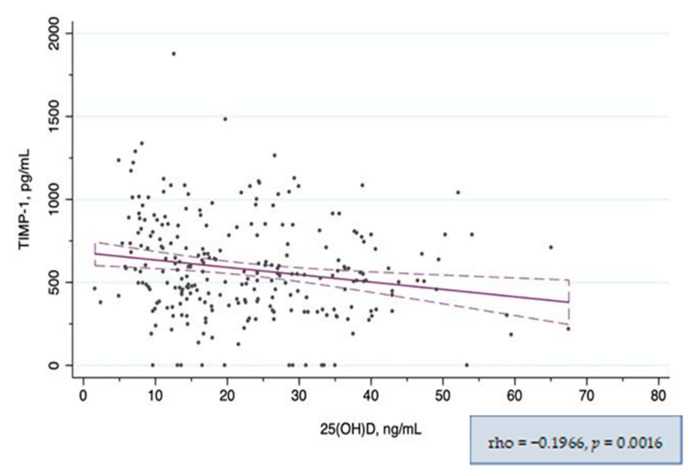
Correlation between serum TIMP-1 and 25(OH)D_3_ levels.

**Table 1 nutrients-14-03484-t001:** Description of the patients’ characteristics according to the vitD_3_ status.

	All Patients	Normal >30 ng/mL	Insufficiency 20–30 ng/mL	Deficiency <20 ng/mL
*n* (%)	256 (100.0)	66 (25.8)	63 (24.6)	127 (49.6)
Age, years	67 (60; 74)	67 (62; 72)	68 (57; 75)	67 (59; 76)
Diabetes duration, years	10 (5; 16)	10 (5; 15)	7 (3; 13)	11 (7; 19)
Sex, % male/female	73/27	50/50	73/27	84/16
BMI, kg/m^2^	28.5 (25.7; 31.1)	27.3 (25.2; 30.0)	29.1 (26.9; 31.5)	28.9 (25.9; 31.3)
CKD, %	48.4	30.3	46.0	59.1
Hypertension, %	76.2	71.2	68.2	82.7
CVD, %	37.1	40.9	27.0	40.2
Smokers, %				
Current	18.0	15.1	20.6	18.1
Former	28.5	28.8	30.2	27.6
Non-smoker	53.5	56.1	49.2	54.3
Serum creatinine, mg/dL	1.0 (0.8; 1.3)	0.8 (0.7; 1.1)	0.9 (0.7; 1.2)	1.1 (0.8; 1.6)
Serum cystatin, mg/dL	1.01 (0.81; 1.45)	0.87 (0.77; 1.13)	1.01 (0.82; 1.40)	1.10 (0.86; 1.74)
eGFRcr, mL/min/1.73 m^2^				
2009 CKD-EPI	78 (47; 94)	83 (67; 94)	86 (50; 96)	69 (41; 88)
2021 CKD-EPI	83 (50; 98)	88 (72; 99)	90 (53; 101)	73 (45; 93)
Urea, mg/dL	42 (32; 60)	40 (30; 50)	38 (32; 52)	46 (34; 78)
Urate, mg/dL (*n* = 217)	5.7 (4.7; 6.8)	5.3 (4.6; 6.3)	5.9 (5.1; 6.9)	5.8 (4.6; 7.0)
Plasma glucose, mg/dL	130 (108; 158)	122 (102; 142)	128 (104; 161)	134 (112; 167)
Serum HbA1c, %	6.6 (6.0; 7.4)	6.3 (5.8; 6.9)	6.7 (6.0; 7.4)	6.8 (6.1; 7.7)
Total cholesterol, mg/dL	156 (132; 178)	160 (136; 182)	152 (132; 178)	152 (131; 177)
HDL, mg/dL	47 (38; 57)	54 (43; 64)	47 (39; 55)	45 (36; 54)
LDL, mg/dL	82 (61; 98)	79 (62; 100)	84 (60; 101)	81 (61; 96)
Triglycerides, mg/dL	108 (80; 147)	96 (73; 128)	97 (76; 138)	117 (86; 175)
Hb, g/dL	13.9 (1.6)	14.1 (1.3)	14.2 (1.6)	13.6 (1.6)
Calcium, mg/dL (*n* = 201)	9.4 (9.0; 9.8)	9.4 (9.1; 9.8)	9.6 (9.1; 9.8)	9.3 (8.9; 9.7)
Albumin, g/dL (*n* = 202)	4.2 (3.9; 4.5)	4.3 (4.0; 4.7)	4.3 (4.0; 4.6)	4.1 (3.8; 4.4)
Calcium_corr_, mg/dL (*n* = 191)	9.2 (8.7; 9.7)	9.2 (8.8; 9.6)	9.2 (8.9; 9.7)	9.2 (8.8; 9.7)
Phosphate, mg/dL (*n* = 149)	3.5 (3.1; 3.8)	3.4 (3.1; 3.8)	3.5 (3.1; 3.8)	3.5 (3.1; 3.8)
Intact PTH, pg/mL (*n* = 145)	63.8 (38.7; 132.0)	44.5 (29.7; 70.0)	49.9 (40.0; 97.3)	112.1 (53.7; 201.0)
CRP, mg/dL (*n* = 171)	0.30 (0.10; 1.40)	0.23 (0.10; 0.47)	0.20 (0.12; 0.90)	0.40 (0.10; 2.00)
UACR	18.3 (7.0; 112.2)	10.5 (6.4; 48.0)	14.3 (6.5; 124.0)	24.8 (9.0; 212.0)
Loop diuretics, %	22.3	16.7	20.7	26.0
Thiazides, %	27.0	22.7	20.6	32.3
CCB, %	33.7	27.3	31.8	38.1
RAAS blockage, %	69.9	66.7	63.5	74.8
ACEI	14.1	10.6	11.1	17.3
ARA	41.4	34.9	33.3	48.8
Other	14.4	21.2	19.1	8.8
Paricalcitol, %	7.4	6.1	4.8	9.5
Vitamin D supplement, %	20.3	40.9	28.6	5.5

Data are presented either as median (p25; p75) or as mean (SD). *Note:* if > 10% of values were missing, the number of observations available for each variable is shown in parenthesis. *Abbreviations:* ACEI: angiotensin-converting enzyme inhibitors, ARA: aldosterone receptor antagonists, BMI: body mass index, Calcium_corr_: albumin-corrected calcium, CCB: calcium channel blockers, CKD: chronic kidney disease, CKD-EPI: Chronic Kidney Disease Epidemiology Collaboration, CRP: C reactive protein, CVD: cardiovascular disease, eGFRcr: estimated glomerular filtration rate using creatinine, Hb: hemoglobin, HbA1c: glycated hemoglobin, HDL: high-density lipoprotein, LDL: low-density lipoprotein, PTH: parathyroid hormone, RAAS: renin–angiotensin–aldosterone system, UACR: urinary albumin-creatinine ratio.

**Table 2 nutrients-14-03484-t002:** Comparison of the median serum 25(OH)D_3_, MMP-10 and TIMP-1 levels among the vitD_3_ status subgroups.

	All Patients	Normal >30 ng/mL	Insufficiency 20–30 ng/mL	Deficiency <20 ng/mL	*p*-Value
*n*	256	66	63	127	
25(OH)D_3_ [ng/mL], Median (IQR)	20.3 (12.8; 30.8)	37.9 (34.7; 43.0)	25.3 (23.1; 27.2)	12.8 (9.1; 15.8)	<0.001 *
MMP-10 [pg/mL], Median (IQR)	410 (295; 564)	363 (282; 521)	393 (256; 518)	486 (336; 625)	<0.001 **
TIMP-1 [pg/mL], Median (IQR)	546 (377; 776)	503 (317; 669)	546 (380; 827)	593 (421; 818)	0.034 *

*Abbreviations:* IQR: interquartile range; MMP-10: matrix metalloproteinase 10; TIMP-1: tissue inhibitor of matrix metalloproteinase 1. * Kruskal–Wallis test. ** Analysis of variance for log-MMP-10 (3 group comparison).

**Table 3 nutrients-14-03484-t003:** Correlation between circulating MMP-10 and 25(OH)D_3_ levels.

		N	Spearman’s Rho (CI *)	*p*-Value
Overall		256	−0.25 (−0.36; −0.13)	<**0.001**
Subgroup analysis by vitD_3_ status	Normal > 30 ng/mL	66	−0.16 (−0.39; 0.07)	0.174
	Insufficiency 20–30 ng/mL	63	0.00 (−0.23; 0.24)	0.985
	Deficiency < 20 ng/mL	127	−0.24 (−0.40; −0.07)	**0.005**
Subgroup analysis by CKD	CKD	124	−0.28 (−0.46; −0.11)	**0.001**
	No CKD	132	−0.03 (−0.20; 0.14)	0.746

*Abbreviations:* CI: confidence interval; CKD: chronic kidney disease. * CI for Spearman’s rho coefficient were estimated using bootstrapping method.

**Table 4 nutrients-14-03484-t004:** Correlation between circulating TIMP-1 and 25(OH)D_3_ levels.

		N	Spearman’s Rho (CI *)	*p*-Value
Overall		256	−0.20 (−0.32; −0.08)	**0.001**
Subgroup analysis by vitD_3_ status	>30 ng/mL	66	0.10 (−0.17; 0.37)	0.453
	20–30 ng/mL	63	0.12 (−0.14; 0.38)	0.353
	<20 ng/mL	127	−0.28 (−0.45; −0.12)	**0.001**
Subgroup analysis by CKD	CKD	124	−0.24 (−0.40; −0.07)	**0.005**
		132	−0.02 (−0.20; 0.15)	0.795

*Abbreviations:* CI: confidence interval; CKD: chronic kidney disease. * CI for Spearman’s rho coefficient were estimated using bootstrapping method.

**Table 5 nutrients-14-03484-t005:** Association between serum MMP-10 and 25(OH)D_3_ level.

	Beta * (95% CI)	*p*-Value
Crude model	−128.2 (−194.4; −62.0)	<0.001
Model 1	−55.2 (−106.7; −3.8)	0.035
Model 2	−68.7 (−117.8; −19.7)	0.006

*Abbreviations*: CI: confidence intervals. ***** Beta-coefficient for the change in MMP-10 in patients with 25(OH)D_3_ < 20 ng/mL compared to those with 25(OH)D_3_ ≥ 20 ng/mL. *Model 1 was adjusted for body adiposity percentage estimated by CUN-BAE and eGFR estimated by CKD-EPI 2021. Model 2 is additionally adjusted for smoking (current/former/never smoker) and taking vitD supplement (yes/no)*.

## Data Availability

Data is available on request.

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
