# Peer review of "Association between Circulating Levels of 25-Hydroxyvitamin D3 and Matrix Metalloproteinase-10 (MMP-10) in Patients with Type 2 Diabetes"

_nutrients, 2022, doi:10.3390/nu14173484_

Round 1

Reviewer 1 Report

In this manuscript, Daria Abasheva and collaborators studied how vitD3 status is related to MMP-10 levels in patients with type 2 diabetes (T2D). Overall, the study has many strengths, showing that vitD3 repletion status was an independent predictor of MMP-10 levels in T2DM patients.

Author Response

Thank you for your comments.

The manuscript has been edited following the reviewers’ suggestions.

We have emphasised all the changes using the Track Changes function.

  1. According to the reviewers’ suggestion, we conducted a thorough English revision of the manuscript.

Reviewer 2 Report

This research paper is very clear and informative, even though this study sample is not a big one and has some limitations.

1. Nearly half of the patients numbers were 25OHD<20ng/ml, which is not so high. 

2. Is there any clinical significance of the difference of MMP-10 between the groups?

Author Response

Thank you for your comments.

The manuscript has been edited following the reviewers’ suggestions.

We have emphasised all the changes using the Track Changes function.

  1. Regarding the comment #1 (“Nearly half of the patients numbers were 25OHD<20 ng/ml, which is not so high”): indeed, 50% of patients in our sample had vitamin D3 deficit. Although it might represent a limitation for a precise definition of 25-OH-D3 threshold associated with MMP-10 increase, these values are consistent with other studies of 25-OH-D3 circulating levels in patients with diabetes, as we indicate in the Discussion. Larger studies with a wider range of 25-OH-D3 levels would be needed to better address this issue.
  2. Regarding the comment #2 (“Is there any clinical significance of the difference of MMP-10 between the groups?”): we added a brief description of the clinical relevance of higher MMP-10 values to the Discussion. The relative increase of MMP-10 concentration indicates a higher risk of vascular complications. However, as MMP-10 is not a widespread clinical marker, the interpretation of MMP-10 levels difference in absolute numbers is complicated.
  3. Also, we conducted a through English revision of the manuscript.